# Multiple Sharp Fano Resonances in a Deep-Subwavelength Spherical Hyperbolic Metamaterial Cavity

**DOI:** 10.3390/nano11092301

**Published:** 2021-09-04

**Authors:** Ping Gu, Yuheng Guo, Jing Chen, Zuxing Zhang, Zhendong Yan, Fanxin Liu, Chaojun Tang, Wei Du, Zhuo Chen

**Affiliations:** 1Institute of Advanced Photonics Technology, College of Electronic and Optical Engineering & College of Microelectronics, Nanjing University of Posts and Telecommunications, Nanjing 210023, China; guping@njupt.edu.cn (P.G.); 1220024221@njupt.edu.cn (Y.G.); jchen@njupt.edu.cn (J.C.); zxzhang@njupt.edu.cn (Z.Z.); 2School of Science, Nanjing Forestry University, Nanjing 210037, China; zdyan@njfu.edu.cn; 3School of Science, Zhejiang University of Technology, Hangzhou 310023, China; lfx63@163.com; 4School of Physics Science and Technology, Yangzhou University, Yangzhou 225002, China; wdu@yzu.edu.cn; 5National Laboratory of Solid State Microstructures, Schoolof Physics, Nanjing University, Nanjing 210093, China

**Keywords:** spherical hyperbolic metamaterial cavity, multiple Fano resonances, tunability

## Abstract

We theoretically study the multiple sharp Fano resonances produced by the near-field coupling between the multipolar narrow plasmonic whispering-gallery modes (WGMs) and the broad-sphere plasmon modes supported by a deep-subwavelength spherical hyperbolic metamaterial (HMM) cavity, which is constructed by five alternating silver/dielectric layers wrapping a dielectric nanosphere core. We find that the linewidths of WGMs-induced Fano resonances are as narrow as 7.4–21.7 nm due to the highly localized feature of the electric fields. The near-field coupling strength determined by the resonant energy difference between WGMs and corresponding sphere plasmon modes can lead to the formation of the symmetric-, asymmetric-, and typical Fano lineshapes in the far-field extinction efficiency spectrum. The deep-subwavelength feature of the proposed HMM cavity is verified by the large ratio (~5.5) of the longest resonant wavelength of WGM_1,1_ (1202.1 nm) to the cavity size (diameter: 220 nm). In addition, the resonant wavelengths of multiple Fano resonances can be easily tuned by adjusting the structural/material parameters (the dielectric core radius, the thickness and refractive index of the dielectric layers) of the HMM cavity. The narrow linewidth, multiple, and tunability of the observed Fano resonances, together with the deep-subwavelength feature of the proposed HMM cavity may create potential applications in nanosensors and nanolasers.

## 1. Introduction

With the rapid development of plasmonics in the past decades, the plasmon resonances supported by the metallic nanostructures provide us with a possible strategy for device miniaturization down to the nanometric scale [1]. Meanwhile, the intrinsic Ohm loss of metal materials often leads to a broad spectral linewidth, which further limits the overall performance of the plasmon-based nanodevices [1]. Hence, engineering the plasmon mode with a narrow linewidth in the metallic nanostructures is of crucial importance for improving the device performance. Fano resonance (FR) is a well-known interference phenomenon between a discrete autoionized state and a continuum state, which is typically characterized by an asymmetric lineshape and first discovered in atomic physics [2]. Similarly, the interference arising from the coupling between subradiant “dark” and superradiant “bright” plasmon modes in metallic nanostructures can also lead to the formation of FR [3,4,5,6,7]. FR in plasmonic systems has been generally recognized as an efficient strategy to narrow the linewidths of the plasmon modes [3,8,9], and thus has been widely utilized for improving the performances of FR-based nanodevices, such as plasmon rulers [10], nanolasers [11], nanosensors [12,13,14,15,16,17,18,19], molecular identifications [20] and so on. Recently, the multiple FRs have also attracted extensive research attention for the further improvement and expansion of functionalities in metamaterials and plasmonic nanostructures [21,22]. Up to now, three main mechanisms have been used to generate multiple FRs. In the first mechanism, the introduction of “symmetry breaking” into the plasmonic nanostructures enables the excitation of multipolar (high-order) dark plasmons, as well as the generation of multiple FRs, which is considered as the main strategy, reported in various plasmonic nanostructures [22,23,24,25,26,27]. The second mechanism for generating multiple FRs is through the coupling between plasmon and waveguide modes in hybrid plasmonic dielectric-waveguide structures [28,29]. This mechanism is capable of achieving the ultra-narrow (high-*Q*) FRs due to the non-radiative nature of the dielectric waveguide modes [29]. More recently, the coupling between the plasmon modes, supported by the metallic nanostructure, and the multipolar (high-order) dark plasmonic modes, supported by the nanostructured graphene in the graphene–metal metamaterial, can also generate multiple FRs (the third mechanism) [30,31]. In the graphene–metal complex nanostructures, the exceptional electrical and optical properties (ultrahigh electron mobility and tunable conductivity) of graphene can enable the active tunability of FRs by external voltage in the terahertz and mid-infrared spectral range [32,33,34].

A noble metal (Au/Ag) nanoshell consisting of a dielectric nanosphere core and a concentric metallic layer, is the most symmetrical (spherical symmetry) plasmonic nanostructure that simultaneously exhibits broad (superradiant) sphere plasmon modes and narrow (subradiant) cavity plasmon modes [35,36]. The near-field coupling between the cavity and sphere plasmon modes in the metallic nanoshell has been theoretically demonstrated to generate narrow FR [37,38]. However, the excitation efficiency of cavity plasmons in the metallic nanoshell is highly dependent on the cavity size. For smaller-sized metallic nanoshells within the electrostatic limit, the cavity plasmons can only be weakly excited due to their weak interaction with the incident light [39,40]. Until more recently, benefitting from the successful fabrication of high-quality large-sized metallic shells based on self-supporting techniques, the multipolar (high-order) sharp cavity plasmons, as well as their induced multiple narrow FRs, have been experimentally demonstrated to become efficiently excited and generated due to the phase retardation effect [41]. This reveals that beyond the “symmetry breaking” effect, the spherical-symmetry plasmonic nanostructures can also generate multiple FRs.

In the present study, we theoretically investigate the generation of multiple sharp FRs in a deep-subwavelength spherical hyperbolic metamaterial (HMM) cavity formed by five alternating silver/dielectric layers wrapping a dielectric nanosphere core. The multiple sharp FRs are demonstrated to arise from the near-field coupling between multipolar narrow plasmonic whispering-gallery modes (WGMs) and broad-sphere plasmon modes supported by the HMM cavity. Furthermore, we also demonstrate that the electric fields of the WGMs are highly concentrated within the different dielectric layers of the HMM cavity, revealing the narrow linewidths (7.4~21.7 nm) of the WGMs-induced multiple FRs. In addition, the symmetric-, asymmetric-, and typical Fano lineshapes of the WGMs displayed on the extinction efficiency spectrum are demonstrated to be determined by the coupling strength between WGMs and sphere plasmon modes. Finally, the resonant wavelengths of the observed multiple FRs can be easily tuned by varying the dielectric core radius, the dielectric layer thickness, and the refractive index of the proposed HMM cavity.

## 2. Methods

Figure 1 schematically shows the spherical metal/dielectric multilayers nanostructures with total radius of *R*, to be investigated. The proposed nanostructure consisted of a dielectric nanosphere core (radius: *r*, refractive index: *n*) coated by five alternating silver/dielectric layers with thicknesses of *s* and *d*, respectively. For discussion simplicity, the refractive indexes of the dielectric layers were kept the same as the dielectric core with a value of *n*. The coordinates were chosen such that their origins were located at the center of the dielectric core, and the electric field *E*_in_, magnetic field *H*_in_, and the wave vector *k* of the incident light were along *x*, *y*, and *z* axes, respectively. The optical properties (extinction, scattering and absorption) of the plane wave interacting with the proposed spherical metal/dielectric multilayers nanostructures could be solved analytically based on the improved recursive algorithm of well-known Mie scattering theory, due to its perfect spherical symmetry (multilayered sphere) [42,43]. In the calculation, the permittivity of the silver was taken from the experimental data reported by Johnson and Christy [44]. The medium outside the cavity was assumed to be air with a refractive index of 1.0.

In the planar (two-dimensional) layered metal–dielectric multilayers structures, the effective dielectric tensor could be described by *ε*_x_, *ε*_y_, and *ε*_z_ in the rectangular coordinate system, where *ε*_x_ = *ε*_y_ = *ε*_‖_, *ε*_z_ = *ε*_⊥_ [45]. When the dielectric tensors satisfied the conditions: *ε*_‖_ > 0 and *ε*_⊥_ < 0, or *ε*_‖_ < 0 and *ε*_⊥_ > 0, the planar metal–dielectric multilayers structures had hyperbolic-typed dispersion [45]. The effective refractive index theory could be further expanded to the three-dimensional (spherical) metal–dielectric multilayers structures. In the spherical coordinate system, the effective dielectric tensor could be described by *ε*_r_, *ε*_θ_, and *ε*_φ_, where *ε*_θ_ = *ε*_φ_ = *ε*_t_. The *ε*_r_ and *ε*_t_ are expressed as the following, Equations [46,47]:*ε*_r_ = *ε*_m_*n*^2^/[*ε*_m_(1 − *f*) + *n*^2^*f*] (1)
*ε*_t_ = *ε*_m_*f* + *n*^2^(1 − *f*)(2)
where *f* is the ratio of total silver volume to the whole cavity volume, *ε*_m_ is the dielectric constant of the silver, and *n* is the refractive index of the dielectric materials. When the dielectric tensors satisfied the conditions: *ε*_r_ > 0 and *ε*_t_ < 0, or *ε*_r_ < 0 and *ε*_t_ > 0, the spherical metal–dielectric multilayers structures also had hyperbolic dispersion [46,47]. From Equation (1), the epsilon near the zero (ENZ) point of *ε*_r_ corresponded to the ENZ point of silver materials (*λ* = 265 nm). From Equation (2), the ENZ point of *ε*_t_ was determined by the following Equation:*ε*_m_ = *n*^2^(1 − 1/*f*) (3)

For the proposed spherical metal–dielectric multilayers structures with different structural (dielectric core radius, dielectric thickness) and material (dielectric refractive index) parameters in this paper, the maximum ENZ wavelength of *ε*_m_ was about 355 nm. As a result, the condition of *ε*_r_ > 0 and *ε*_t_ < 0 was satisfied at the spectral range of 400–1600 nm for the proposed spherical metal–dielectric multilayers structures, demonstrating the hyperbolic dispersion feature of the proposed nanostructures. One of the main features of the hyperbolic dispersion structures was that the effective refractive index could be very large [45,46,47], which could largely decrease the total cavity size and highly concentrated the resonant energy into the nanometric dielectric layers of the deep-subwavelength cavity [46,47].

## 3. Results and Discussion

Figure 2a displays the calculated total extinction efficiency spectrum, *Q*_ext_ (total), of the HMM cavity with the parameters of *r* = 20 nm, *n* = 1.4, *s* = 20 nm, and *d* = 15 nm (total radius: *R* = 110 nm). The *f* factor is calculated as 0.68633. The most obvious spectral feature is that five narrow resonances characterized by the symmetric, asymmetric, and typical Fano lineshapes are observed in the total extinction efficiency spectrum, which is marked as the Fano *I* (1202.1 nm), *II* (737.5 nm), *III* (684.6 nm), *IV* (564.8 nm), and *V* (475.8 nm), from longer wavelength to shorter wavelength, respectively. The ratio of the longest wavelength mode (1202.1 nm) to the HMM cavity size (diameter: 220 nm) could reach a large value of 5.5, revealing the deep-subwavelength feature of the HMM cavity. Decomposing the total spectrum into separated contributions of electric (*a_l_*) and magnetic (*b_l_*) Mie modes (the subscript *l* represents the angular mode number) is one of the main advantages of analytical Mie scattering theory, which can help us to precisely identify the resonant modes and deeply understand the underlying physical mechanism [42,43]. Results of this analysis of the HMM cavity (solid lines), as well as the same-sized solid Ag nanosphere (dashed lines) for *l* = 1, 2, and 3, are shown in Figure 2b (electric) and Figure 2c (magnetic), respectively. It is clearly seen from Figure 2 that the observed multiple narrow FRs are completely derived from the first three-order electric (*a*_1_, *a*_2_, *a*_3_) contributions (Figure 2b). The first three-order magnetic terms (*b*_1_, *b*_2_, *b*_3_) only provide the ultrabroad linewidth and ultralow intensity contributions to the total extinction efficiency spectrum in the displayed wavelength range, as shown in Figure 2c. The spherical HMM cavity can support multipolar sharped plasmonic whispering-gallery modes (WGM*_l,m_*) with the resonant energy highly concentrated within different dielectric layers depending on the mode order number of *m* [47,48]. The five FRs observed in the total extinction efficiency spectrum correspond to the excitations of plasmonic WGMs, which are marked as WGM_1,1_ (Fano *I*), WGM_2,1_ (Fano *II*), WGM_1,2_ (Fano *III*), WGM_3,1_ (Fano *IV*), and WGM_2,2_ (Fano *V*), respectively (see Figure 2a,b). Besides the five narrow FRs observed in the total and decomposed extinction efficiency spectra, a broad and intense extinction peak centered at 629.2 nm is also observed in the solid metal sphere from the contributions of *a*_1_, which is marked as the ${\mathrm{TM}}_{1}^{S}$ (dashed magenta line in Figure 2b).

In the following, we perform the near-field profiles at the selected wavelengths to further understand the multiple FRs by using the analytical Mie scattering theory [42,43]. Figure 3a–e show the spatial distributions of the electric field intensity enhancement (|*E*/*E*_0_|^2^) at *k–E* (*z–x*) plane for the FRs induced by WGM_1,1_ (*λ* = 1202.1 nm), WGM_2,1_ (*λ* = 737.5 nm), WGM_3,1_ (*λ* = 564.8 nm), WGM_1,2_ (*λ* = 684.6 nm), and WGM_2,2_ (*λ* = 475.8 nm), respectively. It is clearly seen from Figure 3a–e that the electric field intensities are all largely enhanced by 215~4380 times, and are tightly concentrated within the different dielectric layers of the proposed spherical HMM cavity. These highly localized features of electric fields reveal a small mode volume and low radiative loss of the plasmonic WGMs, and thus lead to the formation of multiple sharped (high-*Q*) FRs in the far-field spectrum. In addition, the electric field intensity distributions of the FRs induced by WGM_1,1_, WGM_2,1_, and WGM_3,1_ are found to exhibit twofold (Figure 3a), fourfold (Figure 3b), and sixfold (Figure 3c) symmetry features, and are all concentrated within the first dielectric layer (order number: *m* = 1), revealing that these three resonances correspond to the excitations of the electric dipolar (*l* = 1), quadrupolar (*l* = 2), and octupolar (*l* = 3) plasmonic WGMs in the HMM cavity. Similarly, the electric field intensity distributions for WGM_1,2_ and WGM_2,2_ should have twofold, and fourfold symmetry features, and should be mainly concentrated within the second dielectric layer of the spherical HMM cavity (order number: *m* = 2), as displayed in Figure 3d,e, respectively. In addition, the electric fields for ${\mathrm{TM}}_{1}^{S}$ supported by the solid metal sphere show a similar twofold symmetry, and are mainly concentrated at the silver/air interface as shown in Figure 3f, which corresponds to the excitation of the dipolar sphere plasmon mode. This distribution characteristic also reveals the broad spectral feature of ${\mathrm{TM}}_{1}^{S}$ due to large radiative loss (Figure 2b), because the resonant energy is easily radiated into the surrounding medium of air.

To accurately obtain the characteristics of the resonant modes, the calculated extinction efficiency spectrum is fitted in the vicinity of the FR by using a typical Fano formula: *F*(*ε*) = *σ*_bg_ + *σ*_0_ (*q* + *ε*)^2^/(1 + *ε*^2^), where *σ*_bg_ and *σ*_0_ are the background and normalized extinction, ε = 2(*λ* − *λ*_res_)/*Γ* with *λ*_res_ and *Γ* are the resonant wavelength and linewidth of the resonances, and *q* is the so-called Fano parameter, which describes the degree of asymmetry [3]. Figure 4 shows three examples of such a Fano fitting for the three lowest-order WGMs induced FRs with different line profiles, e.g., WGM_1,1_ (asymmetric lineshape, Figure 4a), WGM_2,1_ (symmetric lineshape, Figure 4b), and WGM_1,2_ (Fano lineshape, Figure 4c). It is obviously seen from Figure 4 that the fitted curves (olive lines) are all in good agreement with the theoretical spectra (red lines) for three different lineshapes, revealing a precise fitting to the FRs. It is also seen from Figure 4 that the linewidths as narrow as 7.4 ~ 21.7 nm are theoretically predicted for the observed FRs. It should be pointed out that the multiple FRs generated by the “symmetry breaking” mechanism in plasmonic nanostructures usually have broad linewidths and need precisely controlled interparticle separation (on the order of several tens of nanometers) [22,24,49]. The linewidths of multiple FRs produced by the plasmonic dielectric-waveguide coupling mechanism can be largely decreased to ~2 nm due to the non-radiative feature of the dielectric waveguide modes [29]. However, the features of planar structures, together with the relatively large vertical thickness, (the excitation of waveguide modes requiring the vertical thickness of dielectric layer larger than a critical value) are unfavorable for the device miniaturization [1]. Compared with the above two mechanisms, our proposed spherical HMM cavity simultaneously has the characteristics of small device size (deep-subwavelength for three-dimensional) and the multiple sharped plasmonic WGMs, which are due to the large effective refractive index and the highly localized feature of resonant energy (revealing low radiative loss, Figure 3), respectively [46]. In addition, the Fano parameter (*q*), as small as 0.02, is obtained for WGM_2,1_, revealing the symmetric lineshape feature (Figure 4b). The *q* factor increases to 0.13 for the asymmetric lineshape of WGM_1,1_ (Figure 4a). Especially for the typical Fano lineshape, the *q* factor can reach a relatively large value of 0.84 for WGM_1,2_, as shown in Figure 4c. It should be pointed out that the lineshapes of WGMs displayed on the extinction efficiency spectrum arise from the near-field coupling between the WGMs and the corresponding sphere plasmon modes. For example, the symmetric lineshape of WGM_2,1_ (Figure 4b) and the asymmetric lineshape of WGM_1,1_ (Figure 4a) result from the weak near-field coupling because of a large resonant energy difference between the WGM_2,1_/WGM_1,1_ and the sphere plasmon modes (Figure 2b). Meanwhile, for WGM_1,2_, its resonant wavelength coincides with the extinction peak of ${\mathrm{TM}}_{1}^{S}$ (Figure 2b), and thus the strong near-field coupling between these two modes leads to the formation of a typical Fano lineshape in the extinction efficiency spectrum, as shown in Figure 4c.

Next, we mainly focus on the tunability of the multiple sharped FRs induced by the multipolar plasmonic WGMs in the spherical HMM cavity. As has been demonstrated in Figure 3, the most electric fields of all the WGMs are highly concentrated within the dielectric layers of the HMM cavity, which allows us to tune the resonant wavelengths of the FRs by changing either the refractive index (*n*) or the thickness (*d*) of the dielectric layers. Figure 5 displays the total extinction efficiency spectra of the HMM cavity for different *n* ranging from 1.0 to 1.5 with a fixed core radius of *r* = 20 nm, silver layer thickness of *s* = 20 nm, dielectric layer thickness of *d* = 15 nm, and an *f* = 0.68633. It is clearly seen that all the five FRs blue-shift to a shorter wavelength with the decreasing refractive index (*n*). It is also apparent that the intensities of the FRs induced by WGM_1,1_, WGM_2,1_ and WGM_3,1_ are gradually increased due to the enhancement of near-field coupling, which is because their resonant wavelengths blue-shift to be close to the corresponding sphere plasmon modes with the decrease in *n* (Figure 5).

Figure 6 shows the total extinction efficiency spectra of the HMM cavity for different dielectric layer thicknesses of *d* from 15 nm (a) to 25 nm (b), and 35 nm (c) (other structural/material parameters: *r* = 20 nm, *s* = 20 nm, and *n* = 1.4). The *f* factors are calculated as 0.68633, 0.57419, and 0.49467, respectively. It is interesting that the resonant wavelength of WGM_1,1_ is first blue-shifted, and then red-shifted as the *d* increases from 15 nm to 35 nm (Figure 6). The observed blue-shift effect of the WGM_1,1_ is due to the reduced layer-to-layer coupling for *d* = 25 nm, as displayed in Figure 6b. As has been demonstrated, the strong layer-to-layer coupling can lead to the apparent red-shifting of the WGMs in the HMM cavity when the dielectric layer thickness is smaller than 15 nm [48]. The red-shifting of WGM_1,1_ for *d* = 35 nm is due to the normal size-increased effect (Figure 6c). It is also clearly seen from Figure 6c that a new resonance of WGM_4,1_ is red-shifted into the current spectral range for *d* = 35 nm. In addition, the resonant wavelengths of the multiple FRs can also be tuned by varying the dielectric core radius (*r*) of the HMM cavity as shown in Figure 7. It is found that the WGMs clearly red-shift to longer wavelengths as the *r* increases from 20 nm (*f* = 0.68633) and 25 nm (*f* = 0.68053) to 30 nm (*f* = 0.67448). For *r* = 25 nm (Figure 7b) and *r* = 30 nm (Figure 7c), a new narrow resonance of WGM_3,2_ is also red-shifted into the current wavelength range. Additionally, the resonant positions of WGM_2,1_ and WGM_1,2_ are spectrally overlapped for *r* = 30 nm, as displayed in Figure 7c.

## 4. Conclusions

In conclusion, we have theoretically demonstrated the generation of multiple sharp FRs with linewidths as narrow as 7.4~21.7 nm, arising from the near-field coupling between multipolar narrow plasmonic WGMs and a broad-sphere plasmon mode in a deep sub-wavelength spherical HMM cavity. The near-field coupling strength between the narrow WGMs and the broad-sphere plasmon modes in the HMM cavity can lead to the formation of the symmetric-, asymmetric-, and typical Fano lineshapes in the extinction efficiency spectrum. The ratio of longest resonant wavelength of WGM_1,1_ to the cavity diameter can reach a relatively large value of 5.5, clearly revealing the deep-subwavelength feature of the proposed HMM cavity. In addition, the resonant positions of the multiple sharped FRs can be easily tuned by varying the dielectric core size, the dielectric layer thickness, and the refractive index. Based on the above-demonstrated properties, our proposed spherical HMM cavity could be used for high-performance deep-subwavelength devices, such as nanolasers [11], nanosensors [20], and plasmon rulers [10].

## Figures and Tables

**Figure 1 nanomaterials-11-02301-f001:**
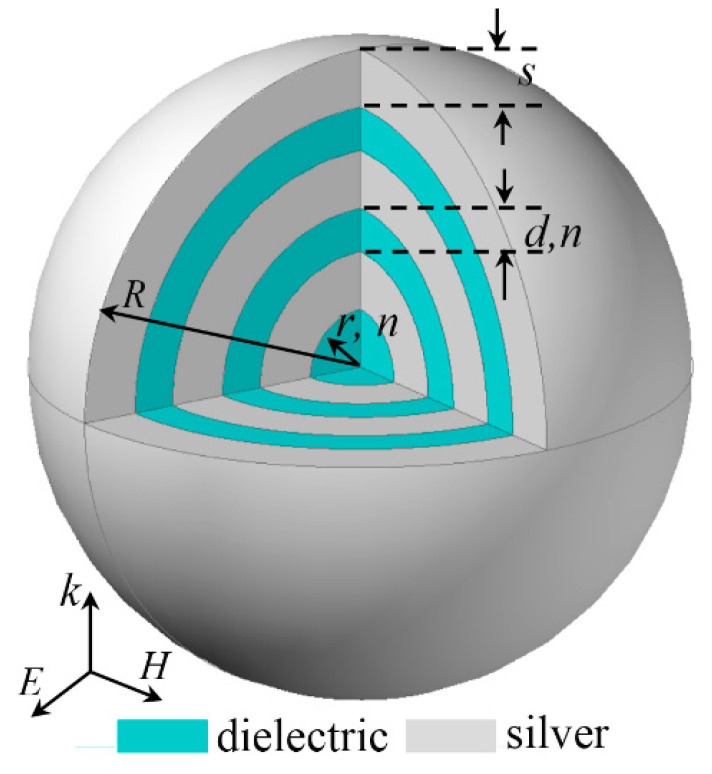
Schematic of a spherical HMM cavity with total radius of *R*, which consists of a dielectric nanosphere core (radius: *r*) and 5 alternating layers of silver (thickness: *s*)/dielectric (thickness: *d*). The refractive indexes of dielectric nanosphere core and the dielectric layers are set as the same value of *n*.

**Figure 2 nanomaterials-11-02301-f002:**
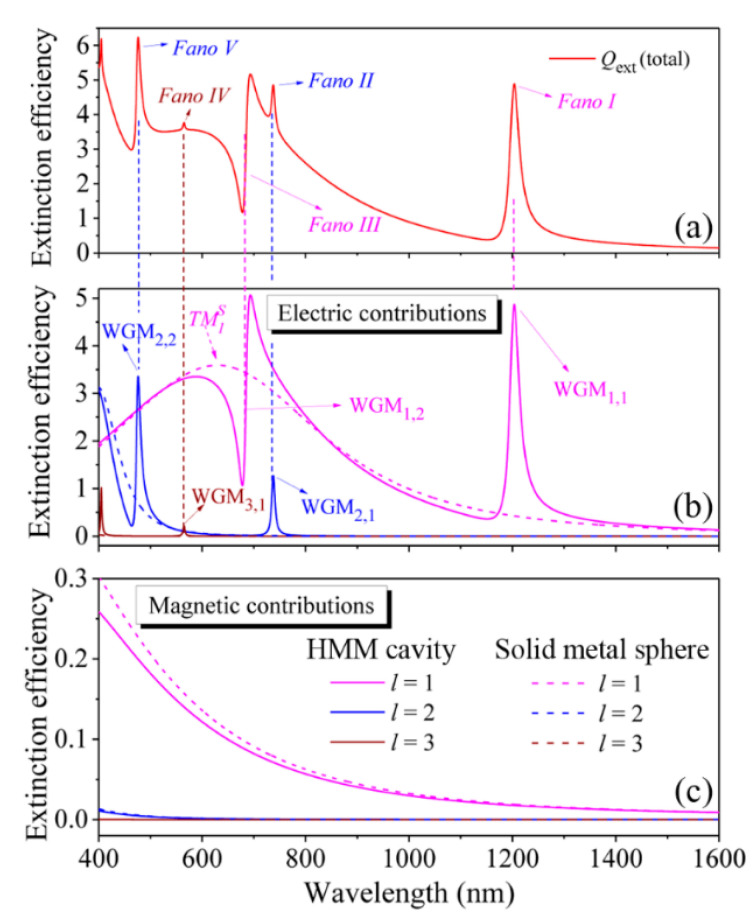
(**a**) The calculated total extinction efficiency spectrum of the HMM cavity with structural/material parameters of *r* = 20 nm, *s* = 20 nm, *d* = 15 nm and *n* = 1.4. The total extinction efficiency spectra of a spherical HMM cavity (solid lines) and a solid Ag nanosphere with radius of 110 nm (dashed lines) are decomposed into separate contributions of the electric (*a_l_*) (**b**) and magnetic (*b_l_*) Mie modes, (**c**) with the angular mode number of *l* = 1,2,3.

**Figure 3 nanomaterials-11-02301-f003:**
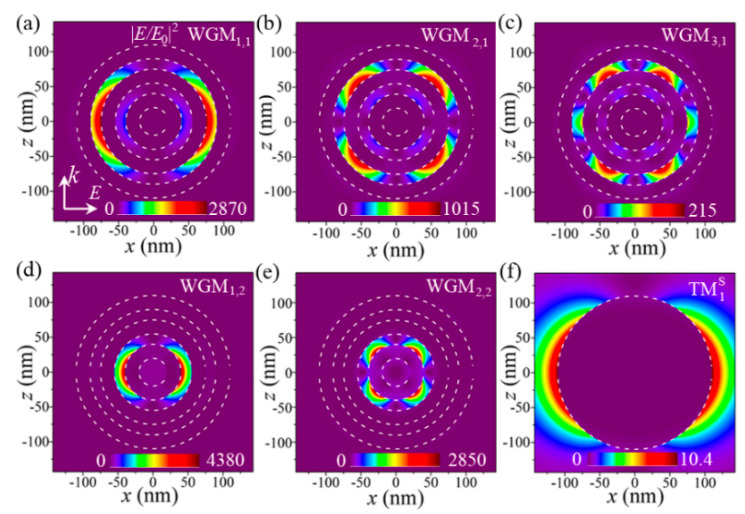
The electric-field intensity (|*E*/*E*_0_|^2^) distributions at *k–E*(*z-x*) plane for WGM_1,1_ (*Fano I*) (**a**), WGM_2,1_ (*Fano II*) (**b**), WGM_3,1_ (*Fano IV*) (**c**), WGM_1,2_ (*Fano III*) (**d**), and WGM_2,2_ (*Fano V*) (**e**) in the spherical HMM cavity and the sphere plasmon mode (${\mathrm{TM}}_{1}^{S}$), supported by a solid Ag nanosphere (**f**), respectively. Dashed circle lines indicate the interfaces of the silver/dielectric or silver/air.

**Figure 4 nanomaterials-11-02301-f004:**
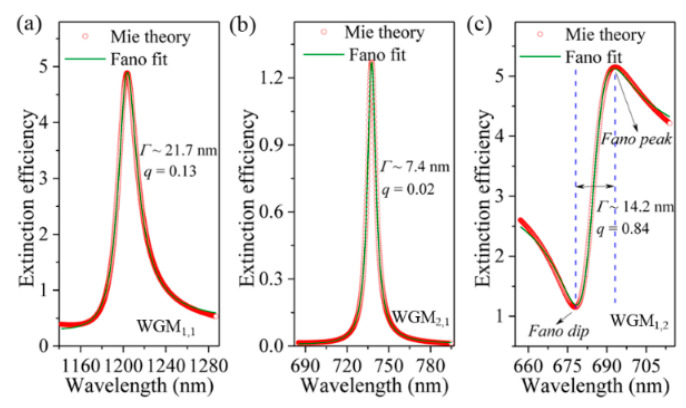
Fano fitting (olive lines) for the FRs induced by WGM_1,1_ (**a**), WGM_2,1_ (**b**) and WGM_1,2_ (**c**), respectively. The red lines display the calculated extinction efficiency spectra.

**Figure 5 nanomaterials-11-02301-f005:**
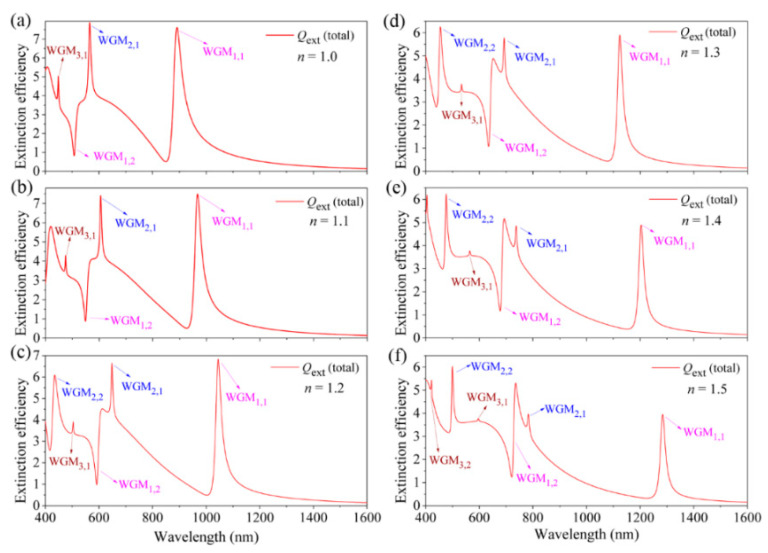
(**a**–**f**) The calculated total extinction efficiency spectra of the spherical HMM cavity for different *n* from 1.0 to 1.5, respectively. Other structural/materials parameters: *r* = 20 nm, *s* = 20 nm and *d* = 15 nm.

**Figure 6 nanomaterials-11-02301-f006:**
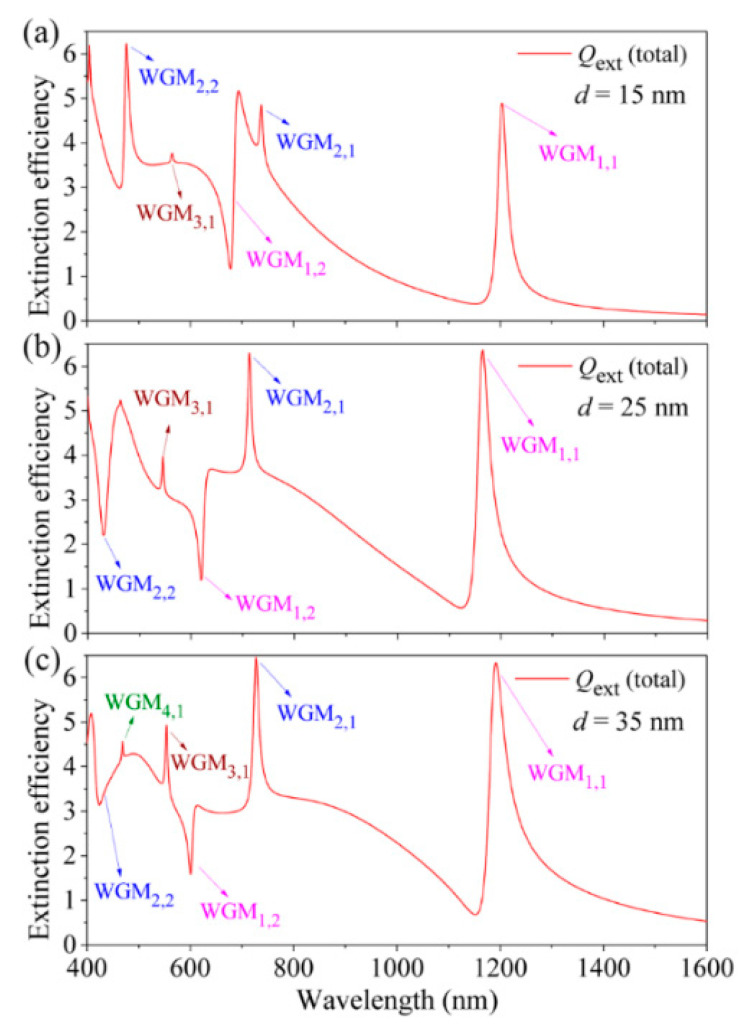
(**a**–**c**) The calculated total extinction efficiency spectra of the spherical HMM cavity for different *d* from 15 nm (**a**) and 25 nm (**b**) to 35 nm (**c**), respectively. Other structural/material parameters: *r* = 20 nm, *s* = 20 nm and *n* = 1.4.

**Figure 7 nanomaterials-11-02301-f007:**
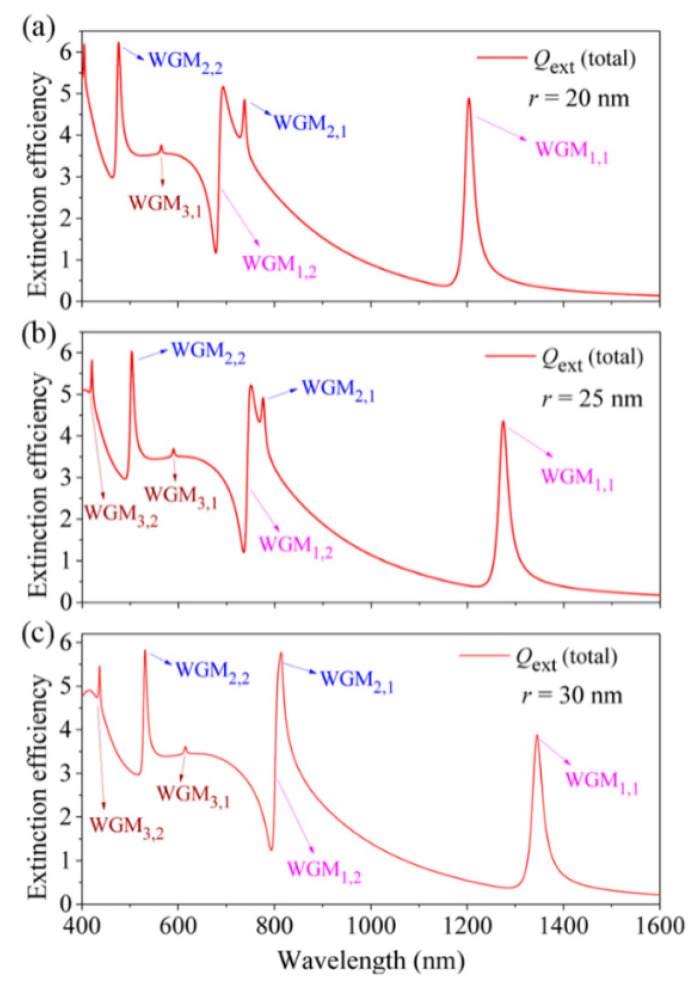
(**a**–**c**) The calculated total extinction efficiency spectra of the spherical HMM cavity for different *r* from 20 nm (**a**) and 25 nm (**b**) to 30 nm (**c**), respectively. Other structural/material parameters: *s* = 20 nm, *d* = 15 nm and *n* = 1.4.

## Data Availability

The study did not report any data.

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
