# Peer review of "Multiple Sharp Fano Resonances in a Deep-Subwavelength Spherical Hyperbolic Metamaterial Cavity"

_nanomaterials, 2021, doi:10.3390/nano11092301_

Round 1

Reviewer 1 Report

The authors present theoretical results for extinction efficiency calculations for spherical metal-dielectric core-shell particles. The methodology is sound and the results are well presented.

As a downside, the paper does not offer much motivation for the results, apart from short literature overview on Fano resonances. Currently it is difficult for me to see the usefulness of the proposed particle design. In my opinion, "Introduction" and perhaps "Conclusions" sections should be improved with this in mind, so that for a reader it would be more clear why are the presented results useful/interesting and what might be the potential applications be. It is also unclear on how the proposed design for multiple FR compares to other available mechanism, and what are the upsides and downsides compared to those.

Some minor notes:

  • In line 65 space is missing before "Until more recently".
  • In line 137 there is space missing after in "Figures 3(a)-3(b)", before "show".
  • In Fig. 3(f) the colorbar is difficult to read.

Reviewer 2 Report

Plasmonic structures attract much attention of researchers nowadays due to their unusual optical properties and perspectives of their applications in nanophotonics. The Manuscript under review presents simulations of the optical properties of a sphere with a multilayered Ag-dielectric shell. The results are promising for possible sensors and light manipulators as ultra-sharp Fano resonance modes are observed in the structure under study. The paper contains interesting numerical results and appropriate discussion on them. I can recommend this Manuscript for the publication in “Nanomaterials” after the following corrections:

  • We authors claim that the multilayered shell under study is a hyperbolic metamaterial but they do not prove this in any way. Keeping in mind the definition of the HMM [A. Poddubny et al, Nature Photonics 7, 948–957 (2013)], I would like to see the spectra of the effective permittivity components for the structure under study or something that proves the evidence of the hyperbolic dispersion regime. Where is the ENZ point? Is the traditional approach for the epsilon components calculation used for the planar multilayered HMM also suitable here?
  • The authors provide the data for different shell parameters. Is the hyperbolic regime achieved for all this parameters?
  • The role of the hyperbolic dispersion in the formation of Fano resonances in the structure is completely not clear. I would like to see the qualitative description of this mechanism in the manuscript.
  • There are some typos as:

line 79: “linshapes”

line 90: “wave factor” – is it “wave vector”?

line 113: “underlining” – I guess, “underlying” is more suitable here.

Round 2

Reviewer 1 Report

No further comments from my side.

Reviewer 2 Report

The authors responded to all my comments and made the necessary changes to the text. I recommend the manuscript for publication